TECHNICAL RELEASE

# Get Free Copy: a multi-repository search platform for biomedical publications

Nodir Kosimkhujaev[1,2] and Kuan-lin Huang[1,2,*]

1 Open Box Science, MA, USA
2 Department of Genetics and Genomic Sciences, Icahn School of Medicine at Mount Sinai, New York, NY, USA

## ABSTRACT

We introduce Get Free Copy (https://getfreecopy.com), a web-based platform designed to streamline the search for biomedical literature across major repositories like arXiv, bioRxiv, medRxiv, and PubMed Central (PMC). Addressing challenges posed by paywalls and fragmented databases, it offers a unified interface for efficient retrieval of free, legitimate copies of biomedical literature. The platform's implementation involves a Node.js backend and dynamic front-end display, enhancing accessibility and research efficiency. As an open-source project, Get Free Copy represents a significant contribution to the open-access movement, inviting global researcher collaboration for further development.

**Subjects** Software and Workflows, Data Mining, Natural Language Processing

## STATEMENT OF NEED

In the rapidly evolving landscape of biomedical research, the ability to access and synthesize relevant literature is critical. The advent of digital publishing has exponentially increased the volume of scientific outputs, presenting both opportunities and challenges for researchers. While the proliferation of academic publications provides more relevant knowledge, such information often stands behind the barriers erected by traditional paywall systems, significantly impeding the free flow of information. These paywalls hinder knowledge exchange and scientific progresses in research, underscoring the importance of open access as a means to democratize knowledge and foster scientific progress [1].

The open access movement has made significant strides in dismantling barriers to research access [2]. Parallel to the development of open-access publishing options, the emergence of preprint servers such as arXiv, bioRxiv, and medRxiv has revolutionized the dissemination of scientific findings, allowing for rapid sharing of research ahead of peer review. The strategic use of these platforms can greatly accelerate scientific communication, facilitating immediate access to research findings [3]. Additionally, PubMed Central (PMC), managed by the National Center for Biotechnology Information (NCBI), is a free digital repository that offers access to full-text biomedical and life sciences journal literature. Despite their growing acceptance, the disparate nature of these repositories complicates literature search, necessitating multiple, often redundant, searches across platforms. Several existing tools, such as Unpaywall [4], LibKey Nomad [5], and EndNote Click (formerly Kopernio) [6], attempt to streamline access to research by providing pathways around paywalls or facilitating easier access to articles. However, these tools often lack

**Submitted:** 11 February 2024

**\*** Corresponding author. E-mail: Kuan-lin.Huang@mssm.edu

Preprint submitted at https://doi.org/10.20944/preprints202404.1595.v1

comprehensive support for preprint publications, are sometimes not universally open, or are restricted to functioning solely as browser or software plugins, limiting their utility across different research contexts for a wide range of researchers.

Building on these foundations, "Get Free Copy" emerges as a pioneering solution designed to bridge the gaps between multiple biomedical literature repositories that provide free versions of research papers. By integrating search results from arXiv, bioRxiv, medRxiv, and PMC, this platform offers a unified interface that simplifies the search and retrieval of research papers. The development of such a tool is not only timely but essential, as researchers navigate the vast and diverse landscape of biomedical literature.

## IMPLEMENTATION

Get Free Copy (https://getfreecopy.com/) is a web-based search engine developed to aggregate and streamline biomedical literature searches across four major repositories: arXiv, bioRxiv, medRxiv, and PMC. This platform significantly enhances the efficiency of locating research papers and preprints by providing a unified interface for results, including key metadata such as the publication title, author names, journal name, publication date, and the digital object identifier (DOI). The Get Free Copy web application is implemented through a Node.js backend ('app.js') that configures a server to handle API (Application Programming Interface) requests and serve static files. It integrates with external scholarly databases and search APIs to fetch academic papers based on user queries. The frontend ('client.js') captures user input from a web form, sends search requests to the backend, and dynamically displays the retrieved academic papers, organizing results by each repository. This dual-component setup allows efficient cross-repository literature searches, enhancing accessibility to academic content.

We demonstrate the Get Free Copy platform's user interface as displayed on a desktop browser and a mobile device (Figure 1). A loading indicator provides visual feedback during query processes. The search term "open science" yields search results from arXiv, bioRxiv, medRxiv and PMC, as one would find by searching directly on each of these repositories (Figure 2). Figures 3–4 further showcases search results returned by searching for a specific author name or institution. For each repository, Get Free Copy show the top query results for titles, author(s), date, journal, and doi in a standardized format. The interface showcases a simple search bar against a clean, white background, allowing users to search across various scientific repositories including arXiv, bioRxiv, medRxiv, and NCBI PMC. The desktop version displays search results from multiple repositories concurrently across its horizontal panels, loading from left to right. The mobile version focuses on user-friendly navigation suited to smaller screens, loading the search results across repositories from top to bottom, where the users can scroll down to see results from other repositories.

## DISCUSSION

The proliferation of scientific literature—coupled with complex paywall systems—has made it challenging to conduct an efficient search of academic publications. Although PMC and preprint servers (e.g., arXiv, bioRxiv, and medRxiv) provide free access to a vast collection of biomedical literature, each operates in isolation, thereby necessitating multiple searches across different databases. This fragmented landscape not only leads to inefficiencies but also exacerbates disparities among researchers lacking access to paid journals.

Our solution, Get Free Copy, addresses these issues by offering a responsive, user-friendly search engine that amalgamates results across repositories. Possible future



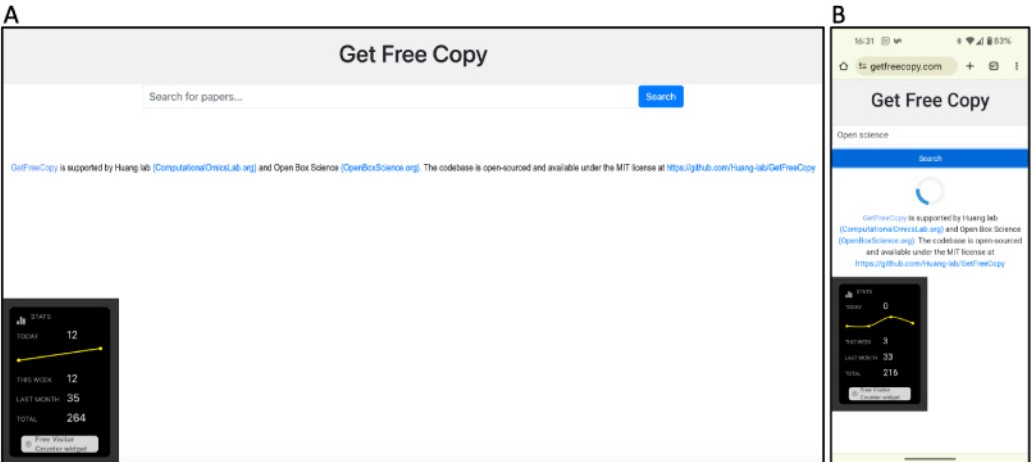

**Figure 1.** The user interface for Get Free Copy, as seen on (A) a desktop browser before query, and (B) a mobile device while loading for queries.

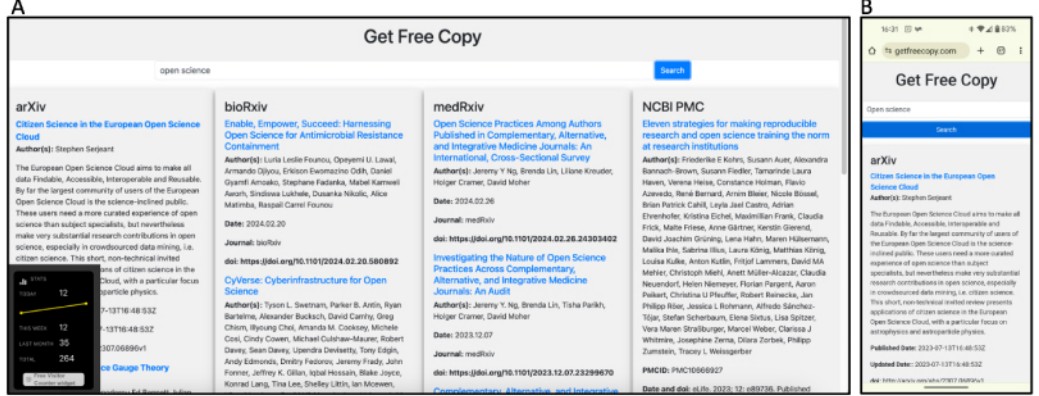

**Figure 2.** Get Free Copy in action, upon querying for the term "open science", as seen on (A) a desktop browser, and (B) a mobile device, demonstrating the platform's responsiveness and multi-repository search capability.

developments of Get Free Copy may focus on expanding the repository coverage for comprehensive literature access and improving user experience with advanced filtering options to streamline the search process. As the user base expands, the platform may also need to be deployed on a server with higher capacities.

Get Free Copy is an open-source project under an MIT license, and we welcome contributions from developers and researchers worldwide to its GitHub repository. Currently, Get Free Copy faces limitations due to the lack of direct search capabilities in the APIs of several preprint servers. We can only rely on web scraping for data extraction, which may lead to server crashes from high query volumes, e.g., 500 Internal Server Errors or yielding no results that likely suggest server overload, rate limiting, or occasional server downtown. This may also result in longer wait time for search queries depending on the response times from each of the servers. We have implemented a loading indicator to enhance user experience by providing visual feedback during the query process. The user

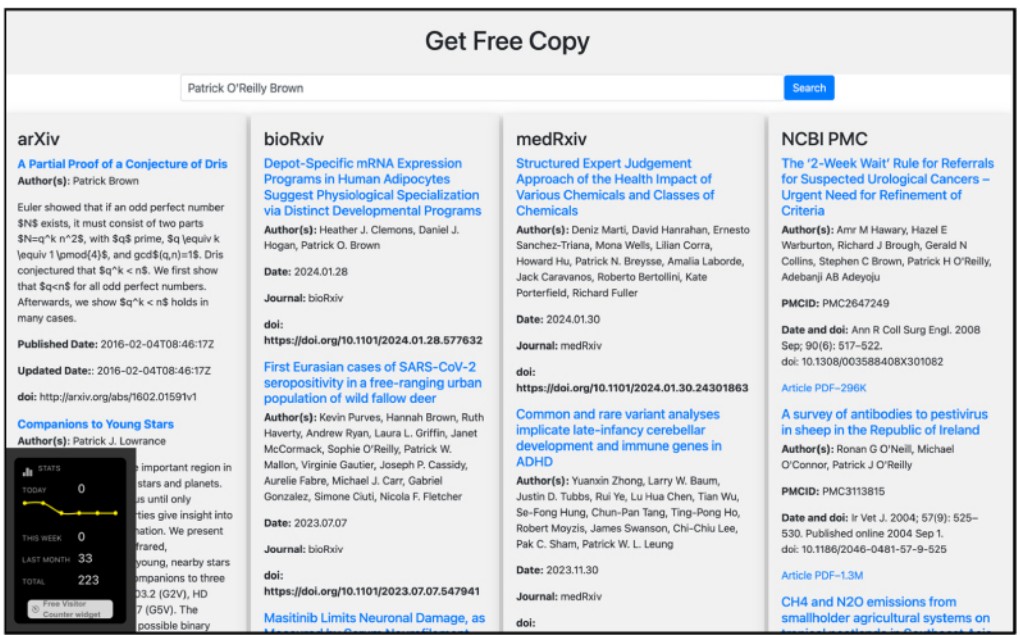

**Figure 3.** Get Free Copy in action, upon querying for the term "Patrick O'Reilly Brown", as seen on a desktop browser.

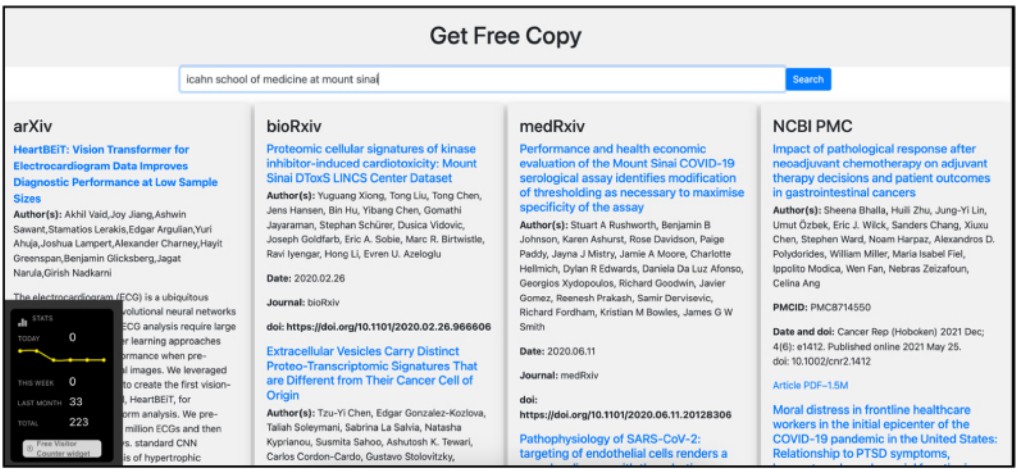

**Figure 4.** Get Free Copy in action, upon querying for the term "icahn school of medicine at mount sinai", as seen on a desktop browser.

experience may be improved through future collaboration with preprint servers and enhanced development.

In summary, we present a search platform—Get Free Copy—that stands at the intersection of open access advocacy, the burgeoning role of preprint servers, and the need for innovative search solutions in biomedical research. By addressing the inefficiencies and disparities in literature access, this platform embodies a significant step forward in making academic research more accessible, efficient, and integrated.

## AVAILABILITY OF SOURCE CODE AND REQUIREMENTS

- Project name: Get Free Copy
- Project home page: https://getfreecopy.com/ and https://github.com/Huang-lab/GetFreeCopy
- Operating system(s): Platform independent
- Programming language: Javascript, CSS, HTML
- Other requirements: none
- License: MIT
- RRID: SCR_025294.

## DATA AVAILABILITY

Get Free Copy is an open-source project, and we welcome contributions from researchers worldwide. Snapshots of the project's code are available under an MIT license at Software Heritage [7], and GigaDB [8].

## ABBREVIATIONS

API: Application Programming Interface; PubMed Central: PMC; National Center for Biotechnology Information: NCBI.

## DECLARATIONS

### Ethics approval and consent to participate

Not applicable.

### Competing interests

KH is a co-founder and board member of a not-for-profit organization, Open Box Science, where he does not receive any compensation. All other authors declare no competing interests.

### Authors' contributions

KH conceived the research. NK and KH developed and deployed the software. KH supervised the study. All authors read, edited, and approved the manuscript.

### Author information

KH: Assistant Professor, Department of Genetics & Genomic Sciences; Associate Member, Tisch Cancer Institute; Faculty Member, Icahn Institute for Data Science and Genomic Technology Icahn School of Medicine at Mount Sinai; Chief Unboxer and Co-Founder, Open Box Science.

### Summary of use: AI tools and technologies

AI-writing tools, specifically ChatGPT [9], in composing and refining drafts of the manuscript and software documentation. GigaDB includes a file, "Summary of Use: AI Tools and Technologies" that includes a key query and output by GPT4 to generate drafts of the manuscript and proposed technical specifications of Get Free Copy [8]. All authors assume full responsibility for the text and software that are generated or refined by these AI tools. All final texts and software codes are extensively modified and verified to ensure the accuracy and integrity of the final work.

## Funding

This work was supported by NIH NIGMS R35GM138113 to KH.

## Acknowledgements

The authors thank members of Open Box Science and the Huang lab for constructive discussion. Large language models (LLM) may have been used in the initial drafts of coding, literature review, and writing of this work. All final codes and texts have been extensively edited and verified by the authors.

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
