## [Editor Report]

Editor’s AssessmentThis software release paper presents Get Free Copy (getfreecopy.com), a web-based platform designed to make it easier to search for biomedical literature across major literature and preprint repositories like arXiv, bioRxiv, medRxiv, and NCBI’s PubMed Central. Providing the Open Source code alongside a webserver that works desktop browsers and mobile devices. Offering a responsive, user-friendly search engine that is a one-stop-shop amalgamating results across repositories. Peer review going through bugs and increasing the background and flagging issues or future developments. This tool potentially aiding open access advocacy, the burgeoning role of preprint servers, and the need for innovative search solutions in biomedical research.Editor’s AssessmentThis software release paper presents Get Free Copy (getfreecopy.com), a web-based platform designed to make it easier to search for biomedical literature across major literature and preprint repositories like arXiv, bioRxiv, medRxiv, and NCBI’s PubMed Central. Providing the Open Source code alongside a webserver that works desktop browsers and mobile devices. Offering a responsive, user-friendly search engine that is a one-stop-shop amalgamating results across repositories. Peer review going through bugs and increasing the background and flagging issues or future developments. This tool potentially aiding open access advocacy, the burgeoning role of preprint servers, and the need for innovative search solutions in biomedical research.

---

## [Reviewer Report]

Reviewer name and names of any other individual's who aided in reviewerPeter EckmannDo you understand and agree to our policy of having open and named reviews, and having your review included with the published manuscript. (If no, please inform the editor that you cannot review this manuscript.)YesIs the language of sufficient quality?YesPlease add additional comments on language quality to clarify if neededIs there a clear statement of need explaining what problems the software is designed to solve and who the target audience is? YesAdditional CommentsIs the source code available, and has an appropriate Open Source Initiative license <a href="https://opensource.org/licenses" target="_blank">(https://opensource.org/licenses)</a> been assigned to the code?YesAdditional CommentsAs Open Source Software are there guidelines on how to contribute, report issues or seek support on the code?YesAdditional CommentsIs the code executable?YesAdditional CommentsThe web server worked for most of the tests I did, but after sending around 10 queries I started getting 500 Internal Server Errors on the requests to biorxiv, medrxiv, and PMC, which showed up as "No results found". Waiting around 20 minutes did not seem to fix the problem. Specifically, the error first started showing up when I searched the phrase "cancer" multiple times, but after it started returning 500 errors they persisted even with different queries.Is installation/deployment sufficiently outlined in the paper and documentation, and does it proceed as outlined?Unable to testAdditional CommentsN/A, because the software is intended to be used on the webIs the documentation provided clear and user friendly?YesAdditional CommentsReadable code with useful commentsAdditional CommentsIs there a clearly-stated list of dependencies, and is the core functionality of the software documented to a satisfactory level?YesAdditional CommentsHave any claims of performance been sufficiently tested and compared to other commonly-used packages? Not applicableAdditional CommentsAdditional CommentsAre there (ideally real world) examples demonstrating use of the software? YesAdditional CommentsIs automated testing used or are there manual steps described so that the functionality of the software can be verified?NoAdditional CommentsAny Additional Overall Comments to the AuthorOverall, the website works well and is a valuable tool to make finding papers easier. I tested a few papers, and it generally worked well (beyond the 500 Internal Server Error mentioned above). After the 500 Internal Server Errors are fixed, I would recommend "Accept" for this paper. A few additional minor comments about the website:  1. arXiv generates DOIs for all papers, but they are not listed (although DOIs are listed for medrxiv, biorxiv, and PMC) 2. Some queries can take a very long time (upwards of 10s). I'm not sure if this is due to the respective websites being slow, but it might improve usability to have some loading indication so users do not get the impression that the website is broken 3. Searching "open science" yields results for all sources, including medRxiv, although in the paper it appears that "open science" yields no results for medRxiv (which seems odd, since one might expect the phrase to be used at least once in a medRxiv paper) 4. DOIs are formatted a little oddly, something like "Doi: doi: ...". Removing the second "doi" would improve clarity. 5. Dates are perhaps formatted incorrectly for medrxiv and biorxiv. I'm seeing the date followed by a lot of random numbers, and I'm not sure what the numbers represent. 6. It might be nice to have a link to the GitHub repository on the website itself, instead of only in the paperRecommendationMinor Revisions

---

## [Reviewer Report]

Reviewer name and names of any other individual's who aided in reviewerJesse XiaoDo you understand and agree to our policy of having open and named reviews, and having your review included with the published manuscript. (If no, please inform the editor that you cannot review this manuscript.)YesIs the language of sufficient quality?YesPlease add additional comments on language quality to clarify if neededIs there a clear statement of need explaining what problems the software is designed to solve and who the target audience is? YesAdditional CommentsIs the source code available, and has an appropriate Open Source Initiative license <a href="https://opensource.org/licenses" target="_blank">(https://opensource.org/licenses)</a> been assigned to the code?YesAdditional CommentsAs Open Source Software are there guidelines on how to contribute, report issues or seek support on the code?YesAdditional CommentsIs the code executable?YesAdditional CommentsIs installation/deployment sufficiently outlined in the paper and documentation, and does it proceed as outlined?YesAdditional CommentsIs the documentation provided clear and user friendly?YesAdditional CommentsIs there enough clear information in the documentation to install, run and test this tool, including information on where to seek help if required?YesAdditional CommentsIs there a clearly-stated list of dependencies, and is the core functionality of the software documented to a satisfactory level?YesAdditional CommentsHave any claims of performance been sufficiently tested and compared to other commonly-used packages? NoAdditional CommentsThe search interface lacks filters and the search speed is slow when has a large number of results.Is test data available, either included with the submission or openly available via cited third party sources (e.g. accession numbers, data DOIs)?YesAdditional CommentsAre there (ideally real world) examples demonstrating use of the software? YesAdditional CommentsIs automated testing used or are there manual steps described so that the functionality of the software can be verified?YesAdditional CommentsAny Additional Overall Comments to the AuthorThis web platform's functions are simple and allow for easy collection and display of search results from various preprint servers. However, it would be beneficial if the author could enhance the search interface by adding filters and advanced search functions. This would enable users to locate articles more conveniently. Additionally, it is important to pay attention to the record format received from preprint servers. e.g. Date: 2023.02.07.527551; Doi: doi: https://doi.org/10.1101/2023.02.07.527551RecommendationMinor Revisions